# NMR resonance assignment and structure prediction of the C-terminal domain of the microtubule end-binding protein 3

**Hazem Abdelkarim**[1], **Ben Hitchinson**[1], **Xinyan Qu**[2], **Avik Banerjee**[3], **Yulia A. Komarova**[2]\*, **Vadim Gaponenko**[1]\*

**1** Department of Biochemistry and Molecular Genetics, College of Medicine, University of Illinois at Chicago, Chicago, IL, United States of America, **2** Department of Pharmacology and the Center for Lung Biology, University of Illinois at Chicago, Chicago, IL, United States of America, **3** Department of Chemistry, University of Illinois, Chicago, IL, United States of America

\* ykomarov@uic.edu (YAK); vadimg@uic.edu (VG)

**Data Availability Statement:** The backbone assignment were deposited in the Biological Magnetic Resonance Databank (http://www.bmrb.

## Abstract

End-binding proteins (EBs) associate with the growing microtubule plus ends to regulate microtubule dynamics as well as the interaction with intracellular structures. EB3 contributes to pathological vascular leakage through interacting with the inositol 1,4,5-trisphosphate receptor 3 (IP$_3$R3), a calcium channel located at the endoplasmic reticulum membrane. The C-terminal domain of EB3 (residues 200–281) is functionally important for this interaction because it contains the effector binding sites, a prerequisite for EB3 activity and specificity. Structural data for this domain is limited. Here, we report the backbone chemical shift assignments for the human EB3 C-terminal domain and computationally explore its ~~EB3~~ conformations. Backbone assignments, along with computational models, will allow future investigation of EB3 structural dynamics, interactions with effectors, and will facilitate the development of novel EB3 inhibitors.

## Introduction

The microtubule (MT) cytoskeleton undergoes rapid remodeling in response to cellular signals, governing cell shape and polarity [1, 2], cell-cell adhesion [3], cell motility and division [4–6], and the spatial organization of intracellular signaling nodes [7, 8]. MT-associated proteins, such as EBs, accumulate at the growing plus ends of MTs and regulate MT dynamics [9–12]. EBs constitute the essential core of the complex of plus-end tracking proteins (+TIPs) [13–17] that establish interactions of MTs with cellular structures [18, 19] and distribute signaling molecules to the cell periphery in a motor-independent manner [20].

In mammals, the EB family consists of three paralogues, EB1, EB2 and EB3, which share a high degree of sequence homology [21]. They are comprised of 260–300 residues organized into the N- and C-terminal domains connected with a variable linker. The N-terminal region presented by the calponin-homology domain binds the MT tip [22], whereas the C-terminal region is required for dimerization [23–25]. Dimerization of EBs is a prerequisite for binding

wisc.edu/) under the BMRB accession code 50003. All other relevant data are within the manuscript.

**Funding:** Y.A.K.; HHSN268201700007C; The National Heart, Lung and Blood Institute, National Institutes of Health, Department of Health and Human Services. H.A.; 132722-PF-18-196-01-DMC, The American Cancer Society V.G.; R01CA188427; The National Institutes of Health (NIH)–National Cancer Institute (NCI) grant A.B.; W81XWH-38817-10509; The Horizon award under the Congressionally Directed Medical Research Program (CDMPRP), Department of Defense (DoD).

**Competing interests:** No authors have competing interests.

to growing MTs as well as interaction with other +TIPs [26–28]. Additionally, the C-terminal region contains the SxIP and LxxPTPh motifs, which are necessary for specific binding of EB partners [24, 29–31], and the EE(Y/F) sequence that is recognized by other cytoskeleton-associated proteins [32–34], including cytoplasmic linker proteins [35], and kinesin [36]. Hence, the C-terminus likely plays a pivotal role in multiple diverse cellular processes.

Despite significant sequence conservation between EBs, they have distinct functions in cells [21, 37, 38]. EBs differ in their expression patterns throughout mammalian tissues and have unique binding partners [7, 21]. EB3, for example, associates with the F-actin-binding protein drebrin and with the E3 ubiquitin ligase SIAH-1, while EB1 and EB2 do not interact with these proteins [39, 40]. Additionally, EB3 but not EB1 interacts with IP$_3$R3 in endothelial cells [38]. Remarkably, genetic ablation of EB3 in endothelial cells protects from pathological vascular leakage and pulmonary edema, suggesting that targeting its function with pharmacological agents might provide a novel strategy for treating inflammatory lung diseases [38]. However, there is little information on EB3 structure to guide drug discovery efforts. Here, we present NMR assignments and *in silico* protein structure prediction of the human EB3 C-terminus (residues 200–281). Our results will provide a structural basis for design of novel EB3 inhibitors.

## Materials and methods

### Protein expression and purification

Preparation of EB3-C-terminus (200–281) with an N-terminal 6X His-tag was performed as described previously [38]. Briefly, the DNA sequence encoding the last 81 amino acids of the EB3 C-terminus was cloned into a pET42a vector and transformed into the BL21 (DE3) strain of *E. coli* (Invitrogen). Bacteria were grown at 37˚C in M9 media containing $^{15}$N and $^{13}$C stable isotopes and 50 μg/ml kanamycin. Protein expression was induced at an OD$_{600}$ of 0.6–0.7, by 250 μM isopropyl 1-thio-β-D-galactopyranoside, after which the cells were cultured at 30˚C for 4 hr. Bacteria were harvested by low-speed centrifugation, and the pellets lysed by sonication in the buffer containing 150 mM NaCl, 5 mM 2-mercaptoethanol, 2 mM CaCl$_2$, 10 mM imidazole, 2 mM phenylmethylsulfonyl fluoride (PMSF), 25 mM Tris HCl, pH 7.4. 6X. His-EB3-C-terminal domain was purified using Ni-NTA beads (Thermo Scientific) equilibrated with 50 column-volumes of binding buffer (25 mM Tris HCl, pH 7.4, 300 mM NaCl, 5 mM 2-mercaptoethanol, 2 mM PMSF). Bacterial lysate (50 ml) was added to the column and the beads were washed with 150 column-volumes of wash buffer (PBS supplemented with 2 mM CaCl$_2$ and the protease inhibitor cocktail (Sigma). After washing, 6X His-EB3-C-terminus was eluted with 150 mM imidazole. Imidazole was removed using a PD-10 desalting column (GE Life Sciences), and concentrated in an Amicon Ultra-15 with 10 kDa cut-off concentrator unit (Millipore, Inc.). The 6X His-tag was cleaved by 1.5% (w/w) recombinant TEV protease at 4˚C for 16 hr. Cleaved EB3-C-terminus was then subjected to gel filtration chromatography over tandem Superdex 200 HR 10/30 columns connected in series and controlled by an AKTA FPLC (GE Life Sciences).

### NMR spectroscopy

HNCO, HNCA, HNCACB, HN(CO)CA, and HN(CO)CACB 3D triple resonance correlation experiments [41] and a 150 ms $^{15}$N-edited NOESY were used for sequential $^1$H/$^{13}$C/$^{15}$N backbone assignment of the EB3 C-terminal domain. All NMR samples were prepared in buffer containing 1X PBS, and 10% D$_2$O (v/v). The final protein concentration was 0.35 mM or 1mM. NMR spectra were acquired at 25˚C on a Bruker 800 MHz spectrometer. Spectra were

processed using NMRPipe [42] and analyzed with SPARKY (http://www.cgl.ucsf.edu/home/sparky) [43].

## Results and discussion

Backbone assignments for the human EB3 C-terminal domain (200–281) were obtained using 350 μM uniformly $^{13}$C and $^{15}$N-labeled protein and triple resonance NMR experiments [44]. These data were subsequently deposited in the Biological Magnetic Resonance Databank (http://www.bmrb.wisc.edu/) [45] under the BMRB accession code 50003.

The $^1$H, $^{15}$N-HSQC spectrum of the EB3 C-terminus showed dispersed peaks indicative of a well-folded protein (Fig 1). The signal intensities were not uniform, suggestive of self-association or conformational dynamics in parts of the protein. We assigned 90% of $^{15}$N and $^1$H$^N$ resonances, as well as 89% of $^{13}$Cα, and 54% of $^{13}$Cβ signals. Assignment of all backbone resonances was precluded by inefficient transfers in three-dimensional experiments that were likely affected by undesirable relaxation processes. The glycine resonances in the C-terminal region were assigned based on $^{15}$N-edited NOESY, as no signals for these residues were observed in the three-dimensional resonance assignment experiments.

Secondary structure prediction analysis was performed using the TALOS+ web server (https://spin.niddk.nih.gov/bax/nmrserver/talos/) [46]. The TALOS+ results indicated significant α-helical content in protein regions including residues 202 to 205, 209 to 210, 215 to 225, 227 to 231, 235 to 237, 246 to 247, 254 to 256, and 273 to 274 (Fig 2); the rest of the protein contained loops.

Due to severe loss of signal in our NOESY experiments, we did not observe sufficient numbers of NOEs for NOE-based protein structure determination. Thus, the three-dimensional structure of the C-terminal domain of EB3 was modeled based on the highly homologous structure of the C-terminal domain of EB1 and the TALOS+ secondary structure results, using the iterative threading assembly refined algorithm on of I-TASSER web server (https://zhanglab.ccmb.med.umich.edu/I-TASSER/) [47–49]. Consistent with the TALOS results and based on EB1 structure (PDB ID: 3GJO), five models generated here described the C-terminal domain of EB3 as an arrangement of three helices (Fig 3). Helices 1 (residues 202–237) and 2 (residues 246–256) had a fixed relative orientations, whereas helix 3 (residues 267–274 in models 1, residues 268–274 in model 2, residues 264–271 in model 3, residues 265–270 in model 4,

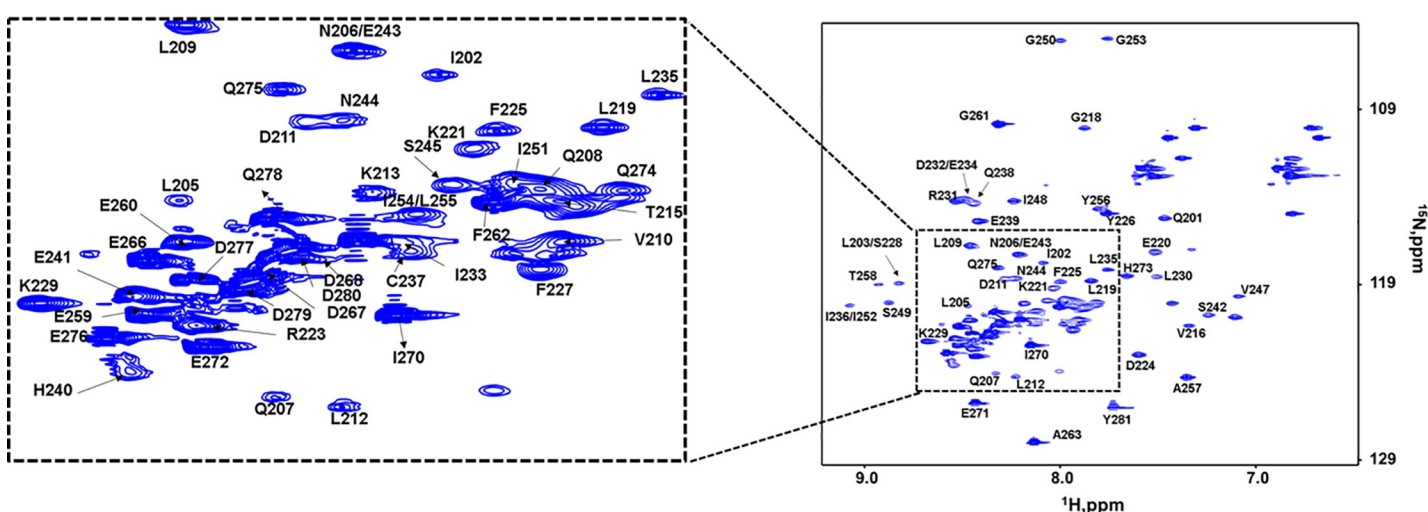

Fig 1. $^1$H, $^{15}$N HSQC spectra of 0.30 mM EB3 C-terminus (200–281). The spectra show assigned well-dispersed signals.

and residues 265–280 in model 5) possessed a variable position and length (**Fig 3**). Further validation by comparing experimental and predicted $^{15}N$ chemical shifts of the five models was made using SHIFTX 2.0 (http://www.shiftx2.ca/) [50]. Using this comparison, we found that

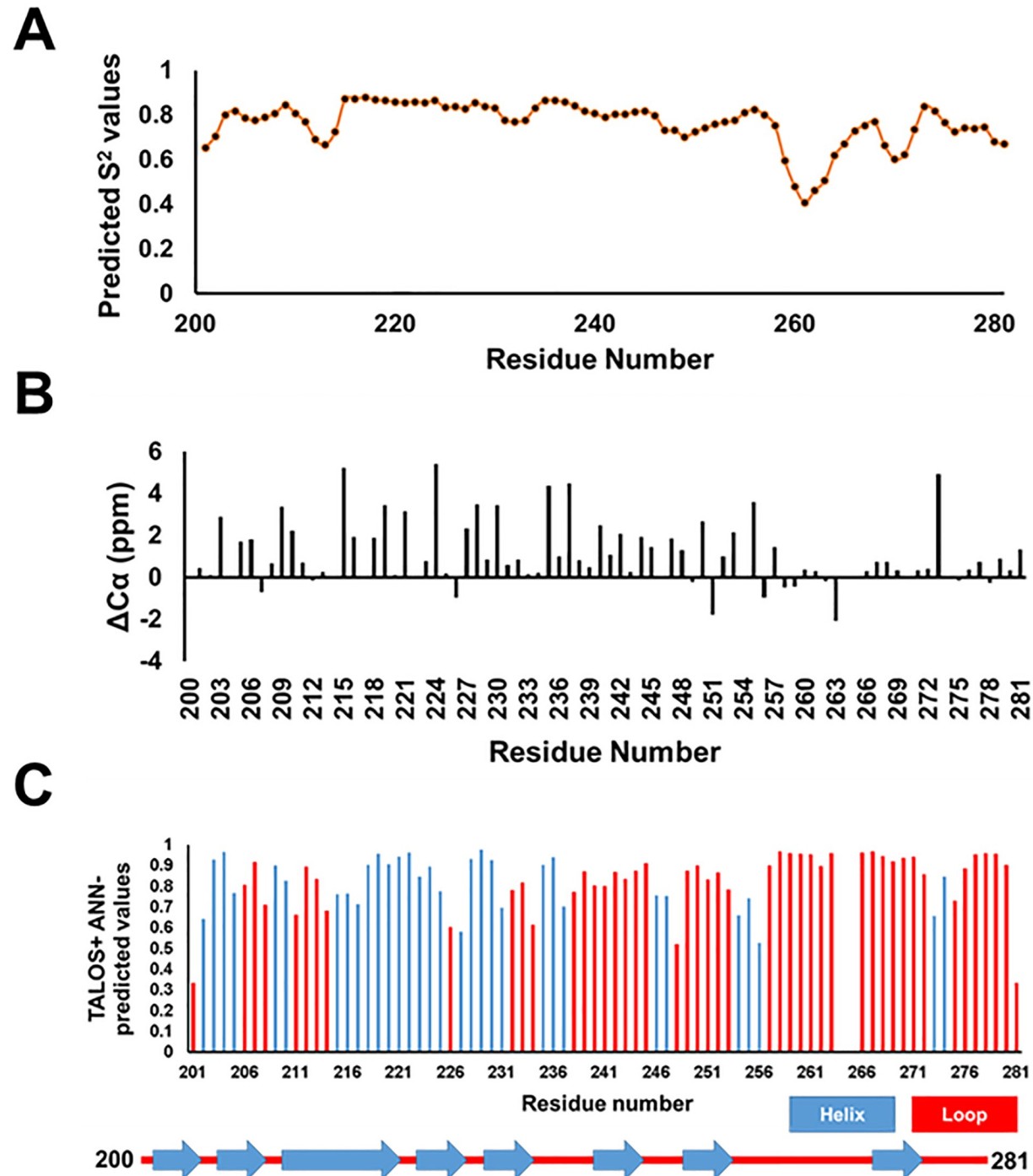

**Fig 2. Secondary structure predictions for EB3 (200–281). A**) Predicted $S^2$ values for the backbone amide groups by the random coil index (RCI) approach indicates varying levels of backbone flexibility. **B**) Deviation from random coil values for Cα chemical shifts indicates the presence of helical elements. **C**) Automated neural network (ANN)-predicted values for the helical region (blue) and loops (red) of the C-terminal domain of EB3. Predicted secondary structure elements are shown using blue arrows for α helices and red lines for coils. RCI $S^2$ and ANN-predicted values were calculated using the TALOS+ web server based on experimental NMR chemical shifts.

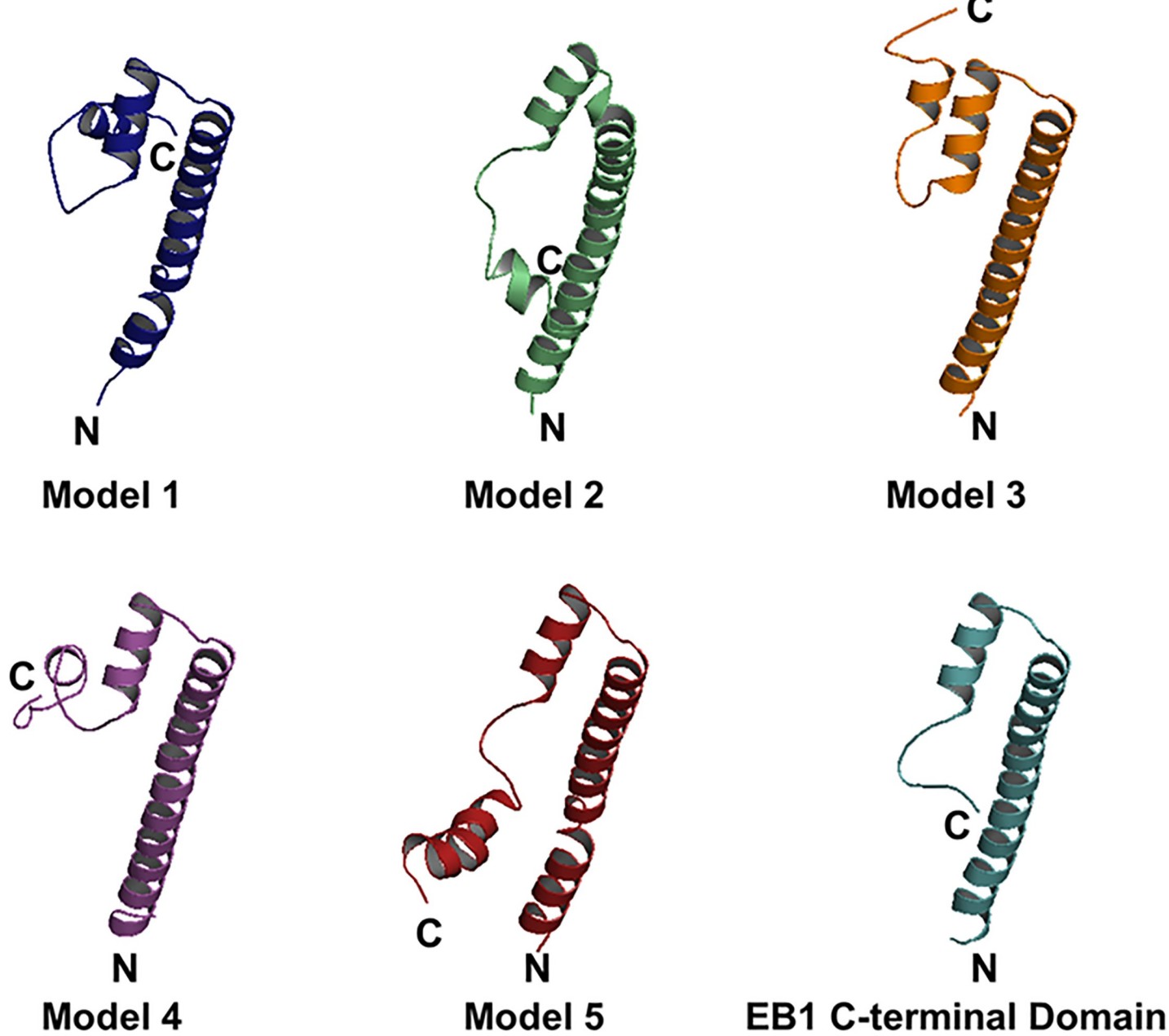

**Fig 3. Computationally predicted structures of the C-terminal domain (200–281) of EB3 and the X-ray structure of C-terminal domain (192–256) of EB1.** Based on secondary structure predictions and the crystal structure of EB1 (PDB ID: 3GJO), five models were generated using the I-TASSER web server. N- and C-termini are marked with N and C, respectively.

model 2 was the most consistent with experimental results presented here (Fig 4). Similar calculations were made for the EB1 crystal structure (PDB ID: 3GJO). The latter showed agreement between the experimentally-derived and predicted $^{15}$N chemical shifts with $R^2$ correlation coefficients of 0.67 and 0.84 for BMRB depositions 34191 and 18371, respectively (Fig 4). Since the C-terminal domains of EB1 (191–268 aa) and EB3 (200–281 aa) share significant amino acid sequence identity of 62.82% as calculated by Protein Blast [21, 51, 52], we generated additional models in I-TASSER based on the structure of EB1 alone (PDB ID: 3GJO)

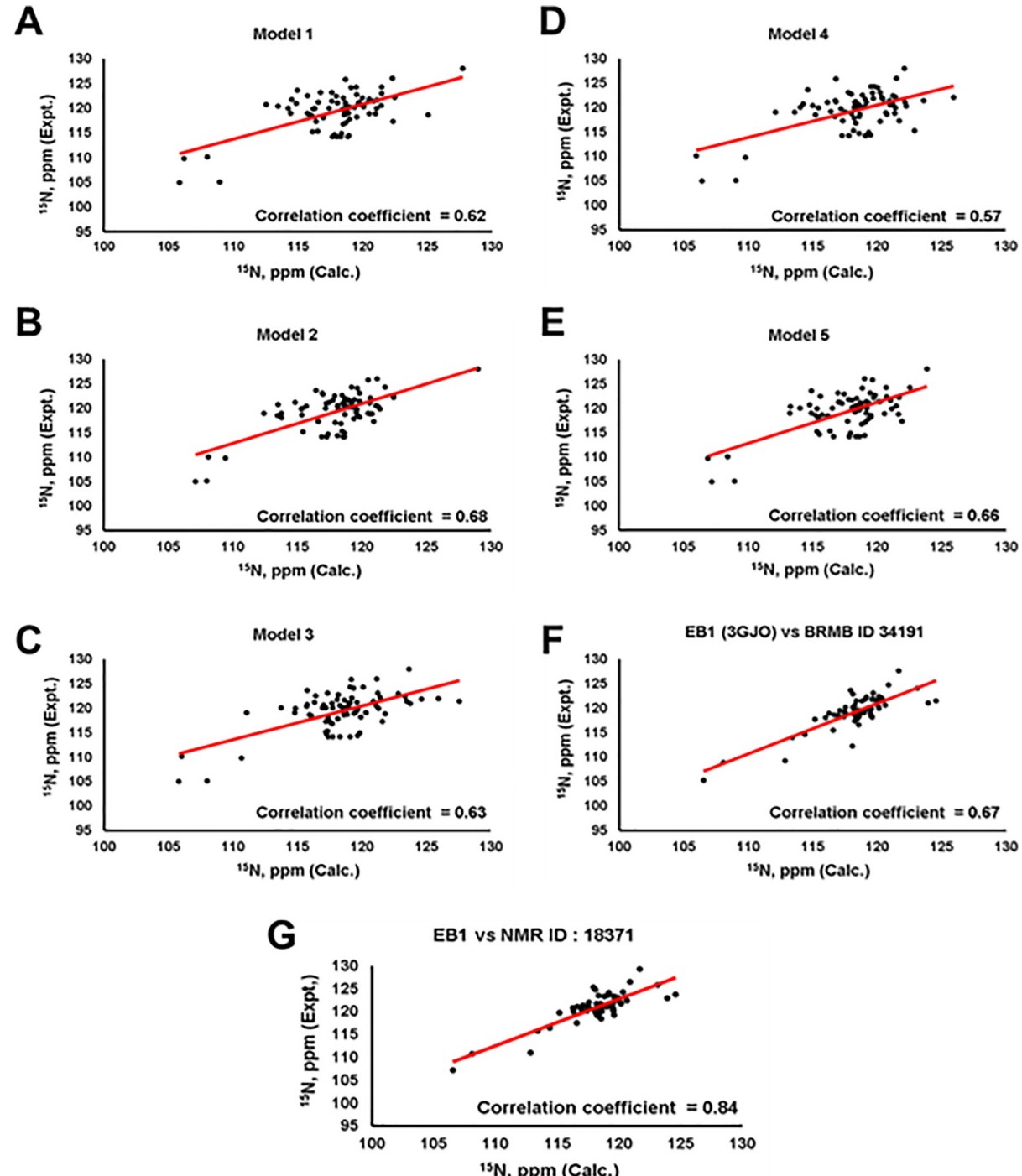

**Fig 4. Validation of *in silico* structure predictions for the C-terminal domain of EB3 based on secondary structure and EB1 (PDB ID: 3GJO) homology restraints, using SHIFTX2. A-E)** Comparisons of experimental $^{15}N$ chemical shifts for the C-terminal domain of EB3 plotted along the Y-axis with the corresponding $^{15}N$ chemical shifts calculated by SHIFTX 2.0 plotted along the X-axis; correlation coefficients are shown for each comparison. Model 2 of the C-terminal domain of EB3 exhibits the highest correlation coefficient, which is comparable to the correlation coefficients for the X-ray structure of the C-terminal domain of EB1 determined using the two sets of $^{15}N$ chemical shift values with BMRB deposition numbers 34191 (**F**) and 18371 (**G**). Hence, Model 2 is a plausible conformation of the C-terminal domain of EB3 in solution.

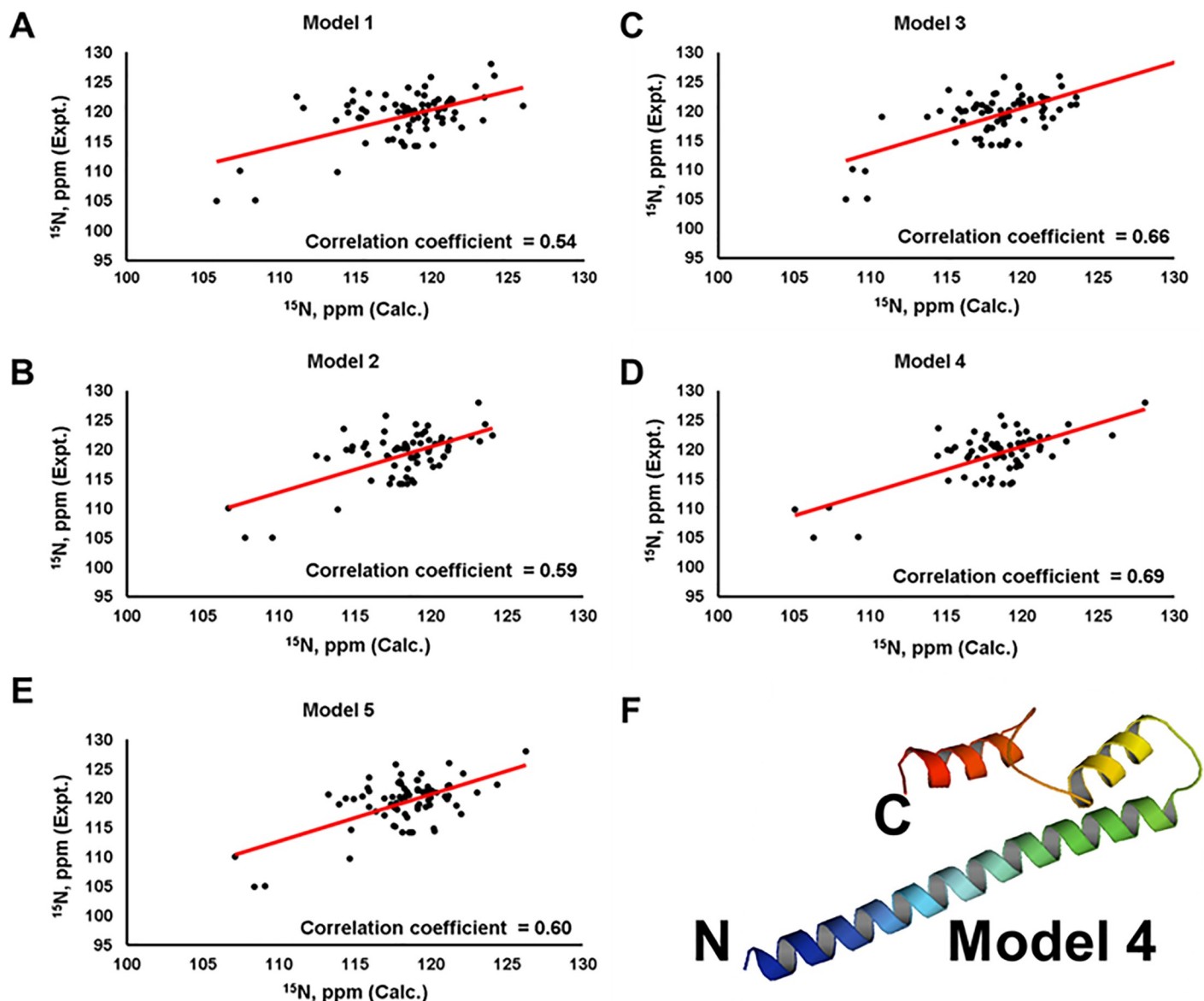

**Fig 5. *In silico* modeling of the C-terminal domain of EB3 based on the structure of EB1 alone (PDB ID: 3GJO) provides an additional plausible conformation.** A-E) Comparisons of experimental $^{15}$N chemical shifts for the C-terminal domain of EB3 plotted along the Y-axis with the corresponding $^{15}$N chemical shifts calculated by SHIFTX 2.0 plotted along the X-axis; correlation coefficients are shown for each comparison. (F) Model 4 exhibits the highest $R^2$ correlation coefficient of 0.69 for experimental versus calculated $^{15}$N chemical shifts. N- and C-termini are marked.

(**Fig 5**), secondary structure restraints alone (**S1 Fig**), or without either EB1 homology or secondary structure information (**S2 Fig**). The best models based on the structure of EB1 with and without NMR-derived secondary structure restraints had comparable correlation coefficients of 0.68 and 0.69 for the predicted versus experimental $^{15}$N chemical shifts, respectively (**Figs 4 & 5**), while removing EB1 homology restraints reduced these correlations (**S1 and S2 Figs**). This suggests that the structure of EB1 is essential for modelling plausible topology of the C-terminal domain of EB3.

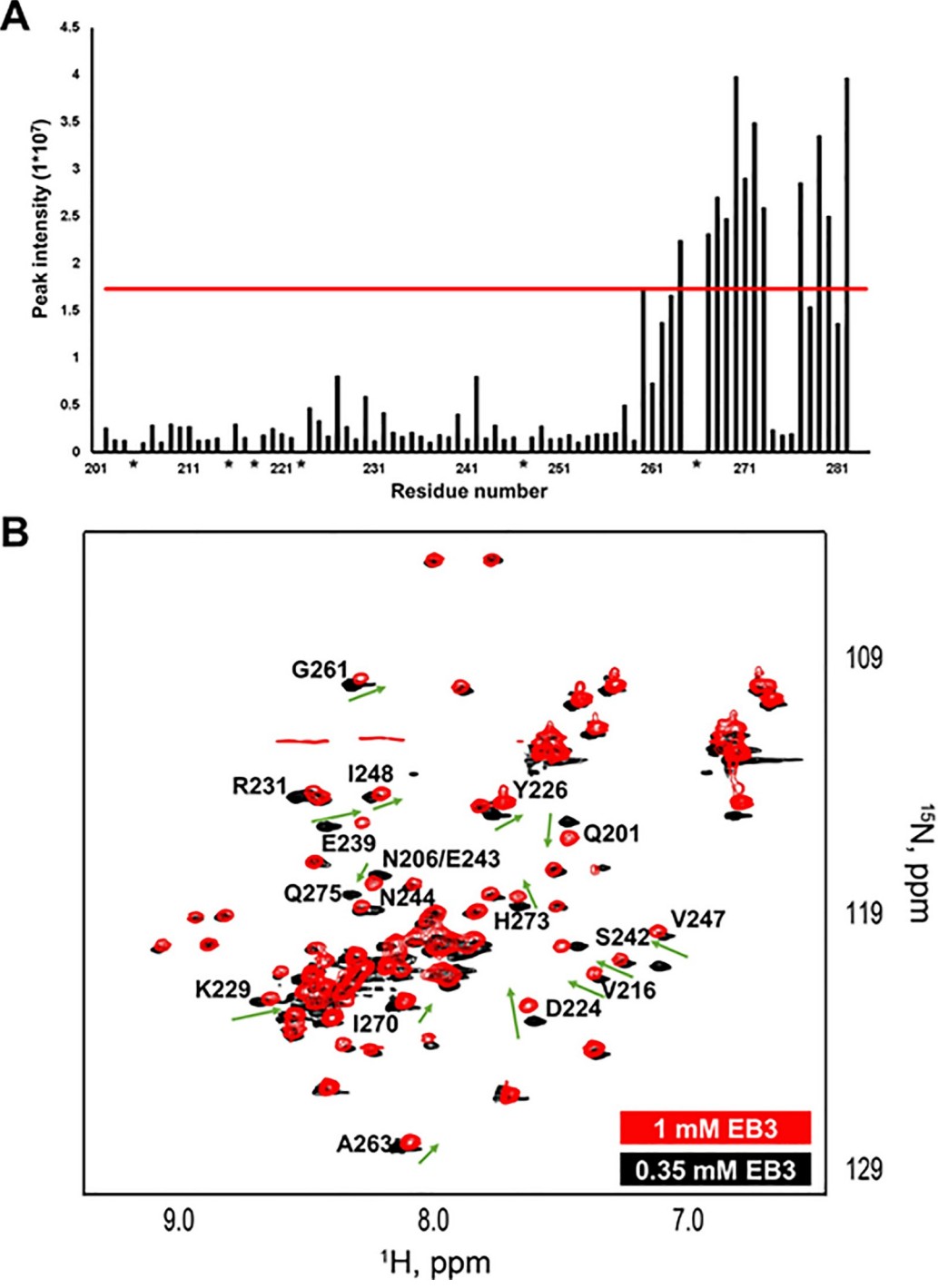

**Fig 6. Dynamic Nature and flexibility of the EB3 C-terminal domain. A)** Analysis of NMR amide signal intensities for the C-terminal domain of EB3. Significant signal intensities observed within the region of 259 to 281 amino acids suggest flexibility of helix 3. The red line represents the peak intensity mean + one standard deviation. Asterisk signs refer to residues that were not assigned. **B)** $^1$H, $^{15}$N overlaid HSQC spectra of the C-terminal domain of EB3 at 0.30 mM (Black) and 1mM (red) show a concentration-dependent chemical shift perturbations (residues Q201; N206; V216; D224; Y226; K229; E239; S242; E243; V247; I248; G261; A263; I270; and H273) and signals broadening (residues R231; N244; and Q275). These changes likely reflect enhanced exchange of EB3 chains at increased protein concentrations. Green arrows show the directionality of the chemical shift change.

Furthermore, analysis of signal intensities in the $^1$H, $^{15}$N HSQC spectrum of the C-terminal domain of EB3 indicated that enhanced relaxation processes might occur in the α-helix 3 region of the protein (**Fig 6A**), suggesting that this region likely samples multiple conformations. For instance, the signal intensities for H273, Q274, and Q275 were low, suggesting increased rigidity in this region of helix 3. Additionally, we have observed concentration dependent changes in the overlaid $^1$H, $^{15}$N HSQC spectrum of EB3 at 0.30 mM and 1mM (**Fig 6B**). These changes involve residues Q201, N206, V216, D224, Y226, K229, R231, E239, S242, E243, N244, V247, I248; G261, A263, I270, H273, and Q275. Residues 201–256 are the part of helix 1 and 2 as well as the flexible loop between these helices in both selected Models (**Figs 4 and 5**). These three regions correspond to the dimeric interface in the C-terminal domain of EB1 (PDB ID: 3GJO). Hence, it is likely that the concentration dependent spectral changes can potentially reflect the chain exchange between EB3 dimers as observed with dimerization of the C-terminal domain of EB1 [25, 26, 31] and EB3 [23, 37].

In summary, we provide assignments for the backbone resonances of the C-terminal domain of EB3. Chemical shift index analysis and molecular modeling suggest that the C-terminal domain of EB3 is highly helical and structurally similar to the C-terminal domain of EB1. The most distal C-terminal portion of EB3 significantly differs from the corresponding portion of EB1 in its amino acid sequence and forms a short helix that likely samples multiple positions relative to α-helices 1 and 2. These models of the C-terminal domain of EB3 can be useful for drug discovery effort.

## Supporting information

**S1 Fig.** *In silico* **structure predictions based on secondary structure restraints show poor agreement with** $^{15}$N **chemical shift data. A-E**) Comparisons of experimental $^{15}$N chemical shifts for the C-terminal domain of EB3 plotted along the Y-axis with the corresponding $^{15}$N chemical shifts calculated by SHIFTX 2.0 plotted along the X-axis; correlation coefficients are shown for each comparison. Models 3 and 5 of the C-terminal domain of EB3 exhibit the highest $R^2$ correlation coefficients of 0.58.
(TIF)

**S2 Fig.** *In silico* **structure predictions without secondary structure and EB1 homology restraints yield low correlations with** $^{15}$N **chemical shift data. A-E**) Comparisons of experimental $^{15}$N chemical shifts for the C-terminal domain of EB3 plotted along the Y-axis with the corresponding $^{15}$N chemical shifts calculated by SHIFTX 2.0 plotted along the X-axis; correlation coefficients are shown for each comparison. Model 2 of the C-terminal domain of EB3 exhibits the highest $R^2$ correlation coefficient of 0.6.
(TIF)

## Acknowledgments

We are grateful to Dr. Bao-Shiang Lee at the University of Illinois Research Resources Center for his help with protein purification.

## Author Contributions

**Conceptualization:** Vadim Gaponenko.

**Data curation:** Yulia A. Komarova, Vadim Gaponenko.

**Formal analysis:** Hazem Abdelkarim, Avik Banerjee.

**Funding acquisition:** Yulia A. Komarova.

**Investigation:** Hazem Abdelkarim, Ben Hitchinson, Xinyan Qu.

**Project administration:** Vadim Gaponenko.

**Supervision:** Yulia A. Komarova, Vadim Gaponenko.

**Validation:** Hazem Abdelkarim.

**Visualization:** Avik Banerjee.

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
