## [Decision Letter · Decision Letter 0]

21 Nov 2019

PONE-D-19-29724

NMR Resonance Assignment and Structure Prediction of the C-Terminal Domain of the Microtubule End-Binding Protein 3

PLOS ONE

Dear Professor Gaponenko,

Thank you for submitting your manuscript to PLOS ONE. After careful consideration, we feel that it has merit but does not fully meet PLOS ONE’s publication criteria as it currently stands. Therefore, we invite you to submit a revised version of the manuscript that addresses the points raised during the review process.

We would appreciate receiving your revised manuscript by Jan 05 2020 11:59PM. To enhance the reproducibility of your results, we recommend that if applicable you deposit your laboratory protocols in protocols.io, where a protocol can be assigned its own identifier (DOI) such that it can be cited independently in the future. For instructions see: http://journals.plos.org/plosone/s/submission-guidelines#loc-laboratory-protocols

We look forward to receiving your revised manuscript.

Kind regards,

Michael Massiah

Academic Editor

PLOS ONE

Journal Requirements:

Reviewers' comments:

Reviewer's Responses to Questions

**Comments to the Author**

1. Is the manuscript technically sound, and do the data support the conclusions?

Reviewer #1: Yes

Reviewer #2: Yes

2. Has the statistical analysis been performed appropriately and rigorously? 

Reviewer #1: I Don't Know

Reviewer #2: I Don't Know

3. Have the authors made all data underlying the findings in their manuscript fully available?

Reviewer #1: Yes

Reviewer #2: Yes

4. Is the manuscript presented in an intelligible fashion and written in standard English?

Reviewer #1: Yes

Reviewer #2: Yes

5. Review Comments to the Author

Reviewer #1: The authors report backbone chemical shift assignments and present in silico structure prediction for the C-terminal domain of the human end binding protein EB3. Overall, the manuscript is clearly written, and the experiments appear to be suitably conducted. The reviewer has several comments (listed below) that should be addressed prior to publication.

Major Comments: -

1. There’s another BMRB file: ID ‘27311’ for NMR assignments of CTD of EB3 (residues 200-281) submitted by the same authors on 2017/11/13. Does the old submission contain incomplete assignments for the same protein fragment? How is the new submission (ID 50003, not available yet on BMRB) different from the old?

2. Figure 4 and page 6: There’s no BMRB file with ID 341191. Please fix to the correct ID: ‘34191’ in figure and text. Also, the R2 correlation coefficients are listed as 0.67 and 0.84 for IDs 18371 and 341191, respectively, in the text. However, these numbers are reversed in the figure. Please correct it.

Minor Comments: -

1. Abstract: Please include abbreviation ‘IP3R3’ for ‘inositol 1,4,5-triphosphate receptor 3, especially since the abbreviation is being directly used by the authors in the introduction afterwards.

2. Figure 1 shows poor resolution in this pdf file. Please check and provide a high-resolution image for the assignments.

3. Figure 2: Please correct the typographical error on the X-axis of all three panels to read ‘Residue Number’.

4. Figure 4: Please correct the typographical error on the X-axis of panels A, C, D, F, and G to read ‘15N, ppm (Calc)’.

5. Figure 5, panel B: Please use uppercase ‘EB3’ instead of lowercase.

Reviewer #2: The manuscript describes partial backbone atoms assignments of cEB3 (200 – 281) using established NMR experiments and modeled structures of cEB3 using available homologous structure of cEB1 supplemented with TALOS+ secondary structural information employing I-TASSER software. I recommend this manuscript for publication provided the authors address the following points:

1) Sequence homology is very high between cEB1 and cEB3 – how different are the calculated structures generated using homology modeling solely based on sequence homology vs model structures reported in the manuscript. This is to show is there any contribution incorporating TALOS+ structural information in structure calculation?

2) The authors observed concentration-dependent chemical shift change for several residues. Are these residues responsible for dimerization?

6. PLOS authors have the option to publish the peer review history of their article (what does this mean?). If published, this will include your full peer review and any attached files.

Reviewer #1: No

Reviewer #2: No

---

## [Author Response · Author response to Decision Letter 0]

22 Mar 2020

Reviewer # 1 (Remarks to Authors)

Minor Comments

5. Figure 5, panel B: Please use uppercase ‘EB3’ instead of lowercase.

Figure 5 is now numbered Figure 6 in the revised manuscript. We have also made sure the uppercase is used for EB3 everywhere in the manuscript.

Reviewer # 2 (Remarks to Authors)

1) Sequence homology is very high between cEB1 and cEB3 – how different are the calculated structures generated using homology modeling solely based on sequence homology vs model structures reported in the manuscript. This is to show is there any contribution incorporating TALOS+ structural information in structure calculation?

We have generated models of cEB3 based on the structure of cEB1 (PDB ID: 3GJO) alone, TALOS+ alone, or without homology and secondary structure restraints. Comparison of experimental and predicted 15N chemical shifts has shown that inclusion of cEB1 homology restraints alone (Fig. 5) or in combination with secondary structure restraints (Fig. 4) leads to the best agreement of the structural models with our NMR experimental data. However, the use of secondary structure restraints alone (S1 Fig.) or excluding secondary structure and cEB1 homology restraints (S2 Fig.) significantly reduced the agreement of structural models with the experimental data. Therefore, the best models could be built either using cEB1 homology restraints alone or in combination with TALOS+-derived secondary structure restraints. 

2) The authors observed concentration-dependent chemical shift change for several residues. Are these residues responsible for dimerization?

We have clarified our statement that the residues exhibiting concentration-dependent chemical shift changes are found in helix 1, helix 2, and the loop region of cEB3. The corresponding regions in the structure of cEB1 (PDB ID: 3GJO) are involved in the homo-dimer interface.

---

## [Decision Letter · Decision Letter 1]

14 Apr 2020

NMR Resonance Assignment and Structure Prediction of the C-Terminal Domain of the Microtubule End-Binding Protein 3

PONE-D-19-29724R1

Dear Dr. Gaponenko,

We are pleased to inform you that your manuscript has been judged scientifically suitable for publication and will be formally accepted for publication once it complies with all outstanding technical requirements.

With kind regards,

Michael Massiah

Academic Editor

PLOS ONE

Additional Editor Comments (optional):

Reviewers' comments:

Reviewer's Responses to Questions

**Comments to the Author**

1. If the authors have adequately addressed your comments raised in a previous round of review and you feel that this manuscript is now acceptable for publication, you may indicate that here to bypass the “Comments to the Author” section, enter your conflict of interest statement in the “Confidential to Editor” section, and submit your "Accept" recommendation.

Reviewer #1: All comments have been addressed

Reviewer #2: All comments have been addressed

2. Is the manuscript technically sound, and do the data support the conclusions?

Reviewer #1: (No Response)

Reviewer #2: Yes

3. Has the statistical analysis been performed appropriately and rigorously? 

Reviewer #1: (No Response)

Reviewer #2: N/A

4. Have the authors made all data underlying the findings in their manuscript fully available?

Reviewer #1: (No Response)

Reviewer #2: Yes

5. Is the manuscript presented in an intelligible fashion and written in standard English?

Reviewer #1: (No Response)

Reviewer #2: Yes

6. Review Comments to the Author

Reviewer #1: (No Response)

Reviewer #2: (No Response)

7. PLOS authors have the option to publish the peer review history of their article (what does this mean?). If published, this will include your full peer review and any attached files.

Reviewer #1: No

Reviewer #2: Yes: TAPAS K MAL

---

## [Editor Report · Acceptance letter]

5 May 2020

PONE-D-19-29724R1 

NMR Resonance Assignment and Structure Prediction of the C-Terminal Domain of the Microtubule End-Binding Protein 3 

Dear Dr. Gaponenko:

I am pleased to inform you that your manuscript has been deemed suitable for publication in PLOS ONE. Congratulations! Your manuscript is now with our production department. 

With kind regards,

on behalf of

Dr. Michael Massiah 

Academic Editor

PLOS ONE